# The effect of induced hyperammonaemia on sleep and melanopsin-mediated pupillary light response in patients with liver cirrhosis: A single-blinded randomized crossover trial

Anna Emilie Kann[1,¤,*], Shakoor Ba-Ali[2,‡], Jakob B. Seidelin[3,‡], Fin Stolze Larsen[1,‡], Steffen Hamann[2,‡], Peter Nissen Bjerring[1]

**1** Department of Intestinal Failure and Liver Diseases, Rigshospitalet, University of Copenhagen, Copenhagen, Denmark, **2** Department of Ophthalmology, Rigshospitalet, University of Copenhagen, Glostrup, Denmark, **3** Department of Gastroenterology, Herlev Hospital, University of Copenhagen, Herlev, Denmark

☯ These authors contributed equally to this work.
¤ Current address: Medical Department, Zealand University Hospital, Koege, Denmark
‡ SB, JBS, FSL and SH also contributed equally to this work.
* annaemiliekann@gmail.com

## Abstract

### Background & aims

Sleep disturbances are related to hepatic encephalopathy and hyperammonaemia in patients with cirrhosis. The circadian rhythm is regulated by light stimulation of the retina via melanopsin-containing ganglion cells. The study aimed to investigate whether induced hyperammonaemia affects the pupillary light response and sleep efficiency in patients with cirrhosis.

### Methods

The study was a single-blinded crossover trial including nine patients with cirrhosis. Sleep was evaluated by Pittsburgh Sleep Quality Index (PSQI) and monitored for twelve nights with wrist accelerometers and sleep diaries. On two experimental days, separated by one week, patients were randomized to ingest either an oral amino acid challenge (AAC) or an isocaloric glucose solution (GS). We measured pupillary light response, capillary ammonia, the Karolinska Sleepiness Scale (KSS), and two neuropsychological tests on both experimental days.

### Results

The patients had poor self-assessed sleep quality. The amino acid challenge led to a significant increase in capillary ammonia and KSS. The time spent in bed sleeping after AAC was longer and with a reduced movement index compared to baseline but not different from GS. We found no difference in the pupillary light response or neuropsychiatric tests when comparing the effect of AAC with GS.

**Data Availability Statement:** All relevant data are within the article and its Supporting information files.

**Funding:** Anna Emilie Kann was supported by a pre-graduate scholarship from the Novo Nordisk Foundation (NNF16OC0020490). The funder had no role in study design, data collection, analysis, manuscript preparation, or decision to publish. No additional funders, either internally or externally.

**Competing interests:** The authors have declared that no competing interests exist.

## Conclusions

Patients with cirrhosis had impaired sleep quality. Induced hyperammonaemia led to increased sleepiness but had no acute effect on pupillary light response or the neuropsychiatric tests.

## Trial registration

Registration number: NCT04771104.

## Introduction

About fifty percent of patients with cirrhosis experience sleep disturbances during life [1]. In previous studies, an association between hepatic encephalopathy (HE) and sleep disturbances has been proposed–e.g., inversion of the sleep-wake cycle and increased daytime sleepiness [2, 3]. The association is thought to be mediated through hyperammonaemia. For instance, Bersagliere et al. observed a parallel increase in subjective sleepiness and plasma ammonia levels [4], and Singh et al. found an improvement in sleep quality due to ammonia-lowering therapy [5]. Furthermore, it has previously been suggested that retinal impairment can be caused by hyperammonaemia and its detoxification by glial cells [6–8].

The circadian rhythm maintains a regular sleep pattern and is partly regulated by light stimulation of the retina via the melanopsin-containing intrinsically photosensitive retinal ganglion cells (ipRGCs). Melanopsin is a photo pigment with maximal sensitivity to light with a wavelength of 480 nm corresponding to blue light. When retina is stimulated with blue light, the ipRGCs detect the light signals and transfer these to higher brain centers, including the suprachiasmatic nucleus (which affects circadian rhythm) and the olivary pretectal nucleus (which affects pupillary light response) [9]. Therefore, the function of ipRGCs can be evaluated by measuring the post-illumination pupillary response with chromatic pupillometry [10].

Based on the above, we proposed a new theory not previously investigated. We hypothesized that hyperammonaemia led to a dysfunction of the ipRGCs and thereby affected sleep regulation in patients with cirrhosis. We conducted a crossover trial to evaluate sleep and the pupillary light response with a pupillometry before and after an oral amino acid challenge. We hypothesized that an oral amino acid challenge induced hyperammonaemia, increased subjective sleepiness, worsened the psychometric tests for minimal encephalopathy, and impaired the pupillary response to blue light stimulation.

## Materials and methods

### Study design

The study was a single-blinded, randomized crossover study consisting of two experimental days separated by one week at the Department of Ophthalmology, Rigshospitalet, Denmark. The assessments on the two experimental days were identical and combined with a randomized intervention of either an oral amino acid challenge (AAC) or an isocaloric glucose solution (GS) followed by five hours of observation. Screening and inclusion of patients were done from 01-01-2017 until 01-03-2018.

**Eligibility criteria.** We included patients with Child-Pugh Class A or B cirrhosis. Patients were excluded if they were <20 years or >80 years of age, had alcohol misuse in the preceding six months, had episodes of hepatic decompensation leading to hospital admissions in the preceding month, or had prior overt HE, or received any HE treatment (lactulose, rifaximin, L-

ornithine L-aspartate or branched-chain amino acid supplementation). Additional exclusion criteria were a history of significant head injury, severe sleep-wake disturbances, neurological/psychiatric comorbidity needing medical treatment, active use of neuroactive drugs or other medication known to affect sleep, traveling across more than two time zones in the preceding three months, shift work in the preceding five years, and diabetes (due to the high prevalence of retinopathy). On the first experimental day, all patients underwent an ophthalmological examination, including refraction, best-corrected visual acuity determination, slit-lamp examination, intraocular pressure measurement (Goldman tonometry), and spectral-domain optical coherence tomography of the macula (Spectralis; Heidelberg Engineering GmbH, Heidelberg, Germany). Inclusion required normal results in all ophthalmologic measurements except hypermetropia and myopia, which were allowed. For this reason, we decided only to assess non-diabetic patients for eligibility to avoid a high rate of screen failures. We also originally planned to include a control group of healthy subjects in the trial protocol, but due to time and resource constraints, we decided to limit the study population to cirrhotic patients only since the most significant treatment effect was expected here.

**Intervention.** With sealed envelopes, the patients were randomized to receive either AAC or GS on the first experimental day and the opposite on the second experimental day. We separated the experimental days by one week to eliminate or reduce the carry-over effect. The patients were blinded for randomization. The AAC consisted of 54 gram banana flavoured amino acid mixture added to 200–300 ml of water. The AAC simulated the composition of hemoglobin in approximately 400 ml of blood. The isocaloric glucose solution contained a similarly flavoured mixture of 72 g glucose added to 200–300 ml of water. Both solutions were ingested orally for 10 minutes [4, 11]. Random allocation sequence, enrolment of patients, and patient assignment to interventions were done by AEK.

## The study assessments

The study assessments were performed at different time points during the inclusion period, as shown in Table 1. The experimental days ran from morning (8.00 am) until afternoon (3.00 pm) to reduce differences in the circadian rhythm between patients.

**Capillary ammonia.** Capillary blood was obtained from the patient's ear lobe at baseline and every hour in the subsequent five hours on each experimental day for ammonia measurements. The ammonia concentration was measured with a blood ammonia analyzer (Pocketchem Device BA, Arkray Factory, Kyoto, Japan). The method is a reliable alternative to arterial measurements. It is based on a micro-diffusion, where the ammonium ions are gasified and cause a colour development proportional to the ammonia concentration in the blood [12].

**Table 1. Study assessments during the inclusion period.**

| Assessments | Before first experimental day | Day 1 (First experimental day) | Day 2 | Day 8 (Second experimental day) | Day 9 |
|---|---|---|---|---|---|
| PSQI | x | | | | |
| Sleep-wake evaluation | x | | x | | x |
| KSS | | 0h, 1h, 2h, 3h, 4h, 5h | | 0h, 1h, 2h, 3h, 4h, 5h | |
| Capillary ammonia | | 0h, 1h, 2h, 3h, 4h, 5h | | 0h, 1h, 2h, 3h, 4h, 5h | |
| PHES | | 0h, 4h | | 0h, 4h | |
| CRT | | 0h, 2h, 4h | | 0h, 2h, 4h | |
| Pupillometry | | 0h, 3h, 5h | | 0h, 3h, 5h | |

PSQI, Pittsburgh Sleep Quality Index; KSS, Karolinska Sleepiness Scale; PHES, psychometric hepatic encephalopathy score; CRT, continuous reaction time

**Sleep evaluation.** Before the first experimental day, the patients completed the validated self-rated assessment tool Pittsburgh Sleep Quality Index (PSQI), which assesses sleep quality and disturbances within the preceding month. The questionnaire generates seven component scores: habitual sleep efficiency, daytime dysfunction, subjective sleep quality, sleep disturbances, sleep latency, sleep duration, and use of sleeping medication. A total score of over 5 indicates poor sleep quality [13].

At baseline and every subsequent hour during the experimental days, the patients reported their level of sleepiness by the Karolinska Sleepiness Scale (KSS). The scale is a 10-point scale going from 1 (extremely alert) to 10 (extremely sleepy, falls asleep all the time) [14, 15].

The patients were equipped with an accelerometer (GT3X+, ActiGraph, Florida, USA) from two days before the first experimental day to two days after the last experimental day. The accelerometer was worn on the wrist of the non-dominant arm during the day and night except when showering. The device records movements in three dimensions to provide data on physical activity and sleep efficiency, i.e., actigraphy. Throughout the study period, the patients completed a sleeping diary containing in-bed time, out-bed time, naps during the day, external factors that could disturb sleep, and un-wear times. The diary data was used for the subsequent data preparation and software analysis (ActiLife 6, Actigraph) [4, 16, 17]. We applied automatic sleep-wake detection of 1-minute epochs using the Cole-Kripke algorithm implemented in the software [17]. The following measures were calculated for each day in the sleep evaluation:

$$Total\ sleep\ time = Total\ time\ asleep(minutes) \tag{1}$$

$$Sleep\ efficiency = \frac{Total\ sleep\ time(minutes)}{Time\ spent\ in\ bed(minutes)} \times 100 \tag{2}$$

$$Movement\ index = \frac{Time\ with\ movements\ during\ sleep(minutes)}{Total\ sleep\ time(minutes)} \times 100 \tag{3}$$

$$Fragmentation\ index = \frac{Number\ of\ periods\ of\ one\ minute\ of\ sleep}{Total\ sleep\ time(minutes)} \times 100 \tag{4}$$

$$Sleep\ fragmentation\ index = Movement\ index + Fragmentation\ index \tag{5}$$

$$Wake\ after\ sleep\ onset = total\ time\ awake\ after\ sleep\ onset(minutes) \tag{6}$$

We defined the baseline sleep measures as an average of two nights preceding both experimental days, i.e., an average of four nights. We compared the baseline sleep to the sleep assessment of the night following each experimental day.

**Neuropsychiatric assessment.** The psychometric hepatic encephalopathy score (PHES) was obtained at baseline and after four hours on each experimental day. PHES is a psychometric paper-pencil test used to diagnose and quantify hepatic encephalopathy [18]. The test is divided into five subtests measuring different skills, e.g., number recognition and motor skill. The skills were scored according to age. The sum of the subtests yields a score from -18 to 6. A score under -4 indicates minimal hepatic encephalopathy.

Continuous Reaction Time (CRT) (EKHO, Aarhus, Denmark) was obtained at baseline and after two hours and four hours on each experimental day. It is an objective measure for hepatic encephalopathy. It consists of a headset and a button connected to a computer. The software generates 150 sound stimuli within different time intervals. The patient must press the button every time the software beeps. The CRT result is an index of all recorded reaction

times, indicating steadiness of the reaction time: $CRTindex = \frac{50^{th}percentile}{90^{th}-10^{th}percentile}$. A result below 1.9 indicates minimal hepatic encephalopathy [19].

**Pupillometry.** We measured the pupillary light response at baseline, after three hours, and after five hours on each experimental day from 8.00 am to 3.00 pm. We used a binocular chromatic pupillometer (DP-2000 Human Laboratory Pupillometer; NeurOptics, Inc., Irvine, CA, USA). In all patients, the right eye was the stimulus eye, while we recorded the pupillary responses of both eyes. Each assessment started with dark adaptation for five minutes, followed by stimulation with red light (633 nm), then five minutes of dark adaption followed by blue light (463 nm). The duration of light stimulation was 20 seconds, and the illuminance level of both light colors was 100 lux. The pupillary size was recorded continuously for 10 seconds before, 20 seconds during, and 60 seconds after light stimulation (Fig 1). The following pupillometric outcomes were calculated [10, 20, 21]:

- The late post-illumination pupillary response (**late PIPR**) equals the mean pupil constriction from 10 to 30 seconds after light stimulation. The late PIPR outcome expresses the activation of the ipRGCs [21].

- The early post-illumination pupillary response (**early PIPR**) equals the mean pupil constriction from start to 10 seconds after light stimulation. The early PIPR expresses the rod photoreceptors, cone photoreceptors, and ipRGCs' response [10, 21].

- Maximal contraction amplitude (**CAmax**) equals maximum pupil constriction 4 to 6 seconds after light stimulation. The CAmax expresses a combined response of rod photoreceptors, cone photoreceptors, and ipRGCs [10, 21].

## Statistical analyses

We used a paired Wilcoxon signed-rank test to compare test results after either AAC or GS. The results were expressed as medians (25% quantile; 75% quantile). The length of the error bars on the plots is the median absolute deviation. P-values<0.05 were considered significant except for sleep evaluation, where we used a Bonferroni corrected p-value (P<0.0167) due to multiple comparisons.

**Outcomes.** The primary outcome was the difference in the late PIPR after AAC compared to GS. The secondary outcomes were differences in arterial ammonia, sleep quality, and measurements for hepatic encephalopathy (PHES and CRT) after AAC compared to GS.

**Sample size.** A power analysis was performed with the "pwr" package for R (v.3.1.0), suggesting a sample size of 15 for paired t-tests (effect size 0.8, significance level 0.05, power 0.8). Unfortunately, due to slow recruitment, we ended the study prematurely after the inclusion of nine patients.

## Ethics

The study was approved by The Committee on Health Research Ethics of the Copenhagen Capital Region, Denmark (Project ID: H15000210). All patients provided written informed consent and were not offered financial compensation. The study was conducted under the Helsinki Declaration.

The study was registered on clinicaltrials.gov. We did not register the study before the initiation of patient enrolment, as it was not a requirement in The Committee on Health Research Ethics for this type of national study with no drug intervention. We have no ongoing or related trials.

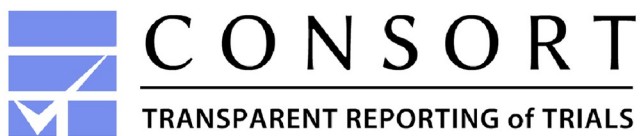

## CONSORT 2010 Flow Diagram

**Enrollment**

Assessed for eligibility (n=15)

Excluded (n= 6)
- ♦ Not meeting inclusion criteria (n= 6)
- ♦ Declined to participate (n= 0)
- ♦ Other reasons (n= 0)

Randomized (n= 9)

**Allocation**

Allocated to intervention at first experimental day (n= 5 )
- ♦ Received allocated intervention (n= 5 )
- ♦ Did not receive allocated intervention (give reasons) (n= 0 )

Allocated to intervention at second experimental day (n= 4)
- ♦ Received allocated intervention (n= 4 )
- ♦ Did not receive allocated intervention (give reasons) (n= 0 )

**Follow-Up**

Lost to follow-up (give reasons) (n= 0)

Discontinued intervention (give reasons) (n= 0 )

Lost to follow-up (give reasons) (n=0 )

Discontinued intervention (give reasons) (n= 0)

**Analysis**

Analysed (n= 5 )
- ♦ Excluded from analysis (give reasons) (n=0 )

Analysed (n= 4 )
- ♦ Excluded from analysis (give reasons) (n= 0 )

**Fig 1. A schematic example of the pupillometry measurement.** The figure shows a normal pupillometry. The mean pupillary diameter was normalized against the mean baseline pupil diameter. The pupils were video recorded for 10 seconds before light stimulation, 20 seconds during light stimulation called "Light ON" (blue arrow), and 40 seconds after light stimulation called "Light OFF" (black arrow). Early PIPR (light grey area) is the early post-illumination pupillary response recorded 0–10 seconds after the light turns off. Late PIPR (dark grey area) is the late post-illumination pupillary response (late PIPR) recorded 10–30 seconds after the light turns off [21].

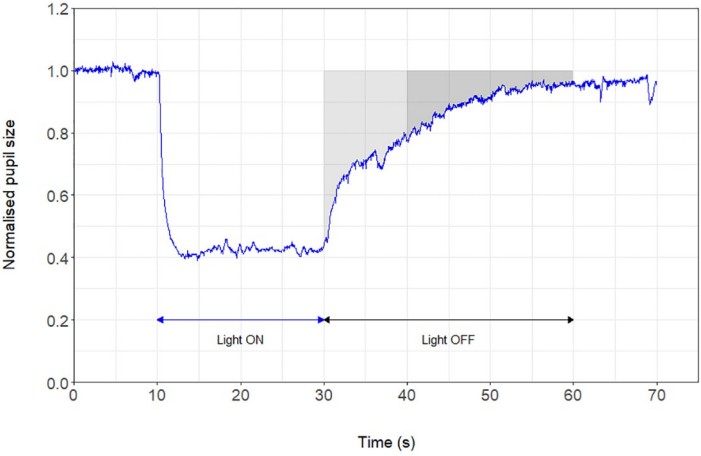

**Fig 2. Flow diagram of enrolment in the study.**

## Results

Fifteen patients were examined for eligibility (see Fig 2). Nine patients were included, and all finished the study according to the study protocol. Of the nine patients, 44% were men, and 67% were classified as a Child-Pugh A. Five patients received AAC on the first experimental day and four on the second experimental day. No critical harms or unintended effects were observed during the study period. The patient characteristics are presented in Table 2.

**Table 2. The characteristics of the included patients with cirrhosis n = 9.**

| Characteristics | No. of patients (%) except when otherwise stated |
|---|---|
| **Men** | 4 (44) |
| **Age,** MEDIAN(RANGE) | 64 (53–73) |
| **BMI,** MEDIAN(RANGE) | 26.2 (20.9–36.7) |
| **Etiology of Liver Cirrhosis** | |
| HEPATITIS B | 1 (11) |
| ALCOHOL | 4 (44) |
| AUTOIMMUNE HEPATITIS | 2 (22) |
| PRIMARY BILIARY CIRRHOSIS | 1 (11) |
| PRIMARY SCLEROSING CHOLANGITIS | 1 (11) |
| **Child-Pugh Score,** MEDIAN(RANGE) | 6 (5–9) |
| **Child-Pugh Classification** | |
| CLASS A | 6 (67) |
| CLASS B | 3 (33) |
| CLASS C | 0 (0) |
| **Randomization**[a] | |
| AMINO ACID CHALLENGE | 5 (56) |
| GLUCOSE SOLUTION | 4 (44) |
| **Self-assessed sleep quality (PSQI)** | |
| NORMAL SLEEP QUALITY (SCORE ≤5) | 3 (33) |
| POOR SLEEP QUALITY (SCORE >5) | 6 (67) |

BMI, Body Mass Index; PSQI, Pittsburgh Sleep Quality Index

[a] Intervention on the first experimental day

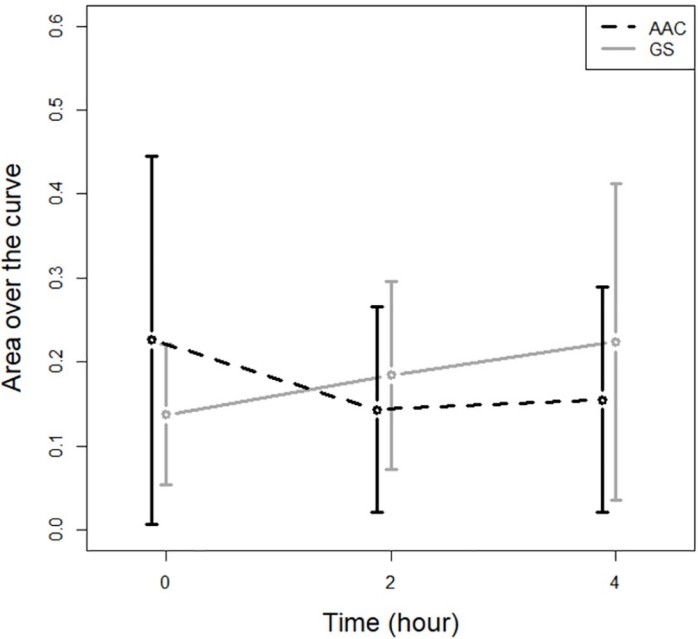

**Fig 3. The late post-illumination pupillary response during the two experimental days.** The late post-illumination pupillary response before, 2, and 4 hours after either amino acid challenge (AAC) or equicaloric glucose solution (GS).

## Pupillometry

We found no significant differences in the late PIPR on AAC day compared to GS day or over time during each day (Fig 3). Additionally, we found no differences in the maximum pupillary contraction or early PIPR on AAC day compared to GS day or over time (S1 and S2 Files).

## Ammonia and KSS

The AAC intervention led to a significant increase in the median capillary ammonia with a maximum of 108 μM (72 μM;130 μM) three hours after intervention (Fig 4A). In addition, the AAC intervention was associated with a significant increase in the median self-rated sleepiness (KSS), with a maximum after 2 hours at a value of 7 (4;8) (Fig 4B). The increase in capillary ammonia and KSS were not seen on GS day.

## Sleep evaluation

The median PSQI score was 6 (3;8). Six of nine patients (67%) had a PSQI score higher than five, indicating poor sleep quality. The sleep assessments by actigraphy showed increased sleep efficiency the night following both GS and AAC experimental day compared to the averaged baseline sleep efficiency (Table 3). Sleep efficiency the night after AAC did not differ from the night after GS. The increased sleep efficiency could mainly be explained by a decrease in the movement index.

## Neuropsychiatric tests

Four of nine patients (44%) had a baseline CRT index lower than 1.9, indicating minimal hepatic encephalopathy. All patients had normal baseline PHES. We did not find significant differences in CRT index or PHES when comparing AAC vs. GS or over time within groups

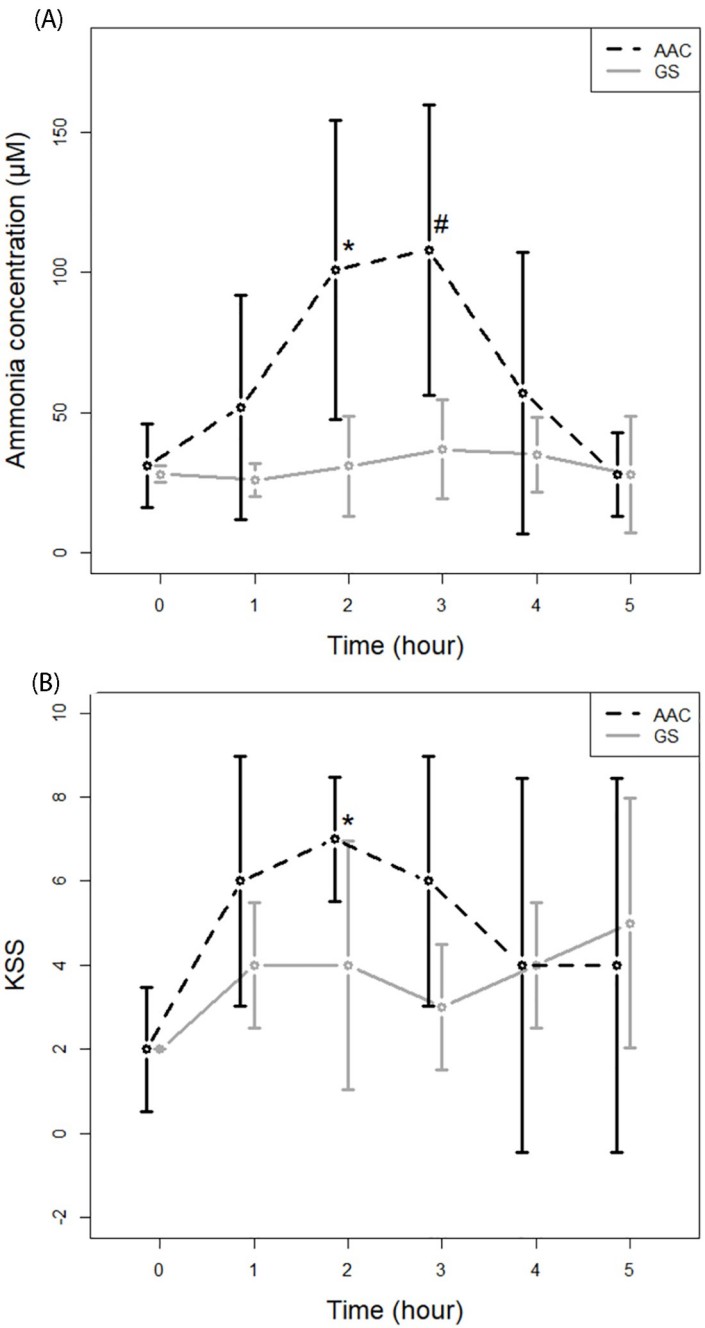

**Fig 4.** A. Capillary ammonia concentration during the two experimental days. Capillary ammonia concentrations (µM) at baseline, 1, 2, 3, 4, and 5 hours after intervention with either an amino acid challenge (AAC) or equicaloric glucose solution (GS). *) Significantly different from GS (p = 0.0156, paired Wilcoxon signed-rank test). #) Significantly different from GS (p = 0.0223). B. Karolinska sleepiness score during the two experimental days. Karolinska sleepiness score (KSS) at baseline, 1, 2, 3, 4, and 5 hours after intervention with either an amino acid challenge (AAC) or equicaloric glucose solution (GS). *) Significantly different from GS (p = 0.0120, paired Wilcoxon signed-rank test).

**Table 3. The average sleep evaluation at baseline compared to the night after amino acid challenge (AAC) and the night after glucose solution (GS) measured by Actigraph.** Median (25% quantile; 75% quantile) n = 9.

| Sleep Parameters | Baseline sleep[I] | Sleep after GS[II] | Sleep after AAC[III] |
|---|---|---|---|
| Sleep Efficiency (%) | 79 (77; 83) | 84 (80; 87)[*] | 87 (83; 90)[**] |
| Total Sleep Time (minutes) | 354 (327; 371) | 390 (381; 426)[*] | 399 (396; 442) |
| Movement Index (%) | 18 (16; 20) | 15 (14; 2310) | 14 (10; 18)[*] |
| Fragmentation Index (%) | 15 (11; 16) | 10 (0; 16) | 9 (5; 13) |
| Sleep Fragmentation Index (%) | 34 (27; 4114) | 28 (23; 328) | 21 (16; 32) |
| Wake After Sleep Onset (minutes) | 79 (73; 98) | 69 (63; 84) | 59 (45; 71)[***] |

AAC, amino acid challenge; GS, glucose solution

[I] Sleep measurements as an average of four nights; two nights before AAC day and two nights before GS.

[II] Sleep measurements the night after GS day

[III] Sleep measurements the night after AAC day

[*]Significantly different from baseline (p = 0.0117, paired Wilcoxon signed rank test)

[**]Significantly different from baseline (p = 0.0039, paired Wilcoxon signed rank test)

[***]Significantly different from baseline (p = 0.0128, paired Wilcoxon signed rank test)

for each experimental day. However, a numerical reduction in the median CRT index following AAC was observed (Table 4).

## Discussion

Our study included nine patients with cirrhosis. Two-thirds had impaired self-reported sleep quality. The AAC intervention was associated with an acute increase in capillary ammonia concentration and subjective sleepiness but had no immediate effect on the melanopsin-mediated pupillary light response or the psychometric tests. We observed more heavy sleep the night after both experimental days compared to baseline sleep but no independent effect of AAC on sleep.

Our hypothesis about impairment of the ipRGC function due to hyperammonemia was not confirmed because the late PIPR was not affected by AAC. Nonetheless, the median late PIPR ranged from 0.14 to 0.23 at the different time points, which is numerically higher than normal values seen in healthy subjects (0.12 95% CI: 0.09–0.15) and lower than in patients with the eye disorder choroideremia (0.30 95% CI: 0.18–0.41) [10]. These comparisons could indicate that patients with cirrhosis might have a slight degree of retinal ganglion cell dysfunction. However, our sample size and lack of a control group of healthy subjects do not allow us to make firm conclusions on baseline PIPR levels in patients with cirrhosis.

**Table 4. Continuous Reaction Time (CRT) Index and the Psychometric Hepatic Encephalopathy Score (PHES) according to randomization and time of examination from intervention.** Median (25% quantile; 75% quantile) n = 9.

| Time from intervention | CRT index | | PHES score | |
|---|---|---|---|---|
| | AAC | GS | AAC | GS |
| **0 hours (baseline)** | 1.973 (1365; 2000) | 1.731 (1464; 2081) | 0 (-1; 1) | -1 (-2; 1) |
| **2 hours** | 1.840 (1535; 1965) | 1.696 (1443; 2185) | - | - |
| **4 hours** | 1.563 (1493; 2343) | 1.714 (1440; 1985) | 2 (-1; 3) | 0 (-2; 2) |

CRT, Continuous Reaction Rime; PHES, Psychometric Hepatic Encephalopathy Score, AAC, AAC, amino acid challenge; GS, glucose solution

The increase in ammonia concentration and sleepiness after AAC was as we suspected and in line with previous studies [4, 11, 22]. We observed more coherent sleep with fewer movements and awakenings the night after AAC and GS, respectively, compared to baseline sleep. We would have expected that sleep after AAC decreased sleep efficiency because Bersagliere et al. [4] found superficial sleep after an equivalent AAC intervention in a similar population. However, differences in the study design could explain the divergence because Bersagliere et al. assessed sleep during a nap in the same afternoon with EEG analysis. Moreover, heavier sleep in our population could be due to exhaustion from a long experimental day rather than the intervention itself.

We found no significant effect of induced hyperammonemia on PHES and CRT, which is in line with previous studies, although slowing reaction time has been reported [4, 11]. Although we did not observe an association between hyperammonaemia and CRT, we found a tendency for a CRT index decline over time on AAC day, reaching values below the index threshold on 1.9. The lack of association between hyperammonemia and PHES could be caused by a potential learning effect of the short period between tests. However, we tried to prevent this by applying four unique PHES test sets per patient.

Plausible explanations of the limited responses we saw—apart from the null hypothesis—are addressed in the following. First, the patients may need a higher peak concentration of ammonia and a longer duration of hyperammonemia to affect the retinal ganglion cells and consequently impair sleep regulation. In addition, the daytime assessment may have affected the results, as ipRGCs are more sensitive to light stimulation in the evening [23]. Second, the composition of the included patients could have affected the results in different ways. For example, the patients' different liver etiology could have contributed to the negative result. Further, the population might have been more susceptible if the patients had more severe liver disease, e.g., Child-Pugh class C, and a history of hepatic encephalopathy. By this, the effect of intervention might have been more pronounced. In the design phase of the study, we decided to use the current inclusion criteria due to a great need for practical cooperation during the time-consuming assessments, especially pupillometry.

A strength of our study was the crossover design with a biologically reasonable wash-out period and the use of non-parametric statistical methods, which gave acceptable protection against type 1 errors and still preserved enough power to detect clinically meaningful effects [24].

## Conclusions

Our study showed that patients with cirrhosis had impaired sleep quality. An oral AAC resulted in an increase in ammonia concentration and associated subjective sleepiness within hours. We found no acute effect of the oral AAC on the melanopsin-mediated pupillary light response; therefore, we could not confirm our main hypothesis.

## Supporting information

**S1 File. First part of original data.** The file contains all collected data except pupillometry data.
(CSV)

**S2 File. Second part of original data.** The file contains all collected pupillometry data. Other data are provided in S1 File.
(CSV)

**S3 File. The trial protocol.**
(PDF)

**S4 File. The consort checklist.**
(DOC)

## Acknowledgments

The authors would like to thank the nurse Bitte Brenholdt Konradsen for her help recruiting patients and the nurse Gitte Marianne Pedersen for her organizational work, which helped facilitate the conduct of the present study.

## Author Contributions

**Conceptualization:** Anna Emilie Kann, Jakob B. Seidelin, Fin Stolze Larsen, Steffen Hamann, Peter Nissen Bjerring.

**Formal analysis:** Anna Emilie Kann, Shakoor Ba-Ali, Peter Nissen Bjerring.

**Investigation:** Anna Emilie Kann, Shakoor Ba-Ali.

**Supervision:** Fin Stolze Larsen, Steffen Hamann, Peter Nissen Bjerring.

**Writing – original draft:** Anna Emilie Kann, Shakoor Ba-Ali, Jakob B. Seidelin, Fin Stolze Larsen, Steffen Hamann, Peter Nissen Bjerring.

**Writing – review & editing:** Anna Emilie Kann, Shakoor Ba-Ali, Jakob B. Seidelin, Fin Stolze Larsen, Steffen Hamann, Peter Nissen Bjerring.

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
