## [Decision Letter · Decision Letter 0]

13 Dec 2021

PONE-D-21-00133The effect of induced hyperammonaemia on sleep and melanopsin-mediated pupillary light response in patients with liver cirrhosis: A single-blinded randomised crossover trialPLOS ONE

Dear Dr. Kann,

Thank you for submitting your manuscript to PLOS ONE. After careful consideration, we feel that it has merit but does not fully meet PLOS ONE’s publication criteria as it currently stands. Therefore, we invite you to submit a revised version of the manuscript that addresses the points raised during the review process.

We look forward to receiving your revised manuscript.

Kind regards,

Jianhong Zhou

Associate Editor

PLOS ONE

Journal Requirements:

2.  Please ensure you have discussed any potential limitations of your study in the Discussion, including study design, sample size and/or potential confounders.

3.  Thank you for submitting your clinical trial to PLOS ONE and for providing the name of the registry and the registration number. The information in the registry entry suggests that your trial was registered after patient recruitment began. PLOS ONE strongly encourages authors to register all trials before recruiting the first participant in a study.

a. your reasons for your delay in registering this study (after enrolment of participants started);

b. confirmation that all related trials are registered by stating: “The authors confirm that all ongoing and related trials for this drug/intervention are registered”.

(The project has been financially supported by a grant from the Novo Nordisk Foundation with grant no. NNF16OC0020490. The funders had no role in study design, data collection and analysis, decision to publish, or preparation of the manuscript.)

Reviewers' comments:

Reviewer's Responses to Questions

**Comments to the Author**

1. Is the manuscript technically sound, and do the data support the conclusions?

Reviewer #1: Yes

Reviewer #2: Yes

Reviewer #3: Yes

2. Has the statistical analysis been performed appropriately and rigorously? 

Reviewer #1: No

Reviewer #2: Yes

Reviewer #3: Yes

3. Have the authors made all data underlying the findings in their manuscript fully available?

Reviewer #1: Yes

Reviewer #2: Yes

Reviewer #3: Yes

4. Is the manuscript presented in an intelligible fashion and written in standard English?

Reviewer #1: Yes

Reviewer #2: Yes

Reviewer #3: Yes

5. Review Comments to the Author

Reviewer #1: The main points of concern in this manuscript are (a) the rather confused description of the known and hypothesized causal relationships and pathways and the light shed on these by this study and (b) the incomplete adherence to standard methods for analysis of cross-over studies.

(a) The introduction explains that liver cirrhosis can cause elevated ammonia level (hyperammonaemia) and hepatic encephalopathy (HE), which in turn can lead to sleep disturbances. The effect on sleep could hypothetically be mediated by damage to retinal cells leading to disturbance of the circadian rhythm. The causal connections and their direction, either known or suspected, between hyperammonaemia, HE and sleep are not clearly described. Ammonia levels were challenged and measures of ammonia level, HE-related variables, eye function and sleep were performed and pair-wise associations between these measures were investigated. It would be useful if the authors could summarize which associations are supported by the study results and what these results tell us about the possible causal pathways.

(b) Analysis of results from a cross-over design should look for a period effect by comparing the results from the first and second periods, since there may be systematic differences due to the subjects’ experience in period 1. It is clear that with an odd number of subjects, the intervention (AAC/GS) is unbalanced with respect to period, which might potentially bias the AAC/GS comparison. With a whole week between periods, we can presumably take the carry-over effect from the first to second period to be negligible – if so, please state this assumption. Further, where appropriate (e.g. PIPR, ammonia level, KSS) the measured values after intervention should be compared with the respective values before intervention (i.e. day 0 of the period), for instance by analyzing the changes (for each subject) during the period.

Minor points

1. Definition of movement index (p.6): why is denominator time spent in bed, rather than TST? As defined, longer TST could lead to increased movement index.

2. Definition of fragmentation index unclear (what is ‘Number of sleep epochs = 1 minute’?).

3. Please justify the assumed effect size (0.8 is unusually large).

4. The distribution of baseline values before AAC an GS respectively should be compared, since similar baselines are need to ensure a fair comparison.

5. In Fig. 3, are mean and standard deviation displayed? With only 9 subjects, it might be possible and more informative to plot the individual series.

6. Table 2: IQR should be given as (lower quartile, upper quartile).

7. Throughout Results, exact p-values should be quoted.

8. The graphic quality of Fig. 2 is very low, to the point of illegibility.

Reviewer #2: This is an interesting, innovative, well presented and honestly discussed piece of research.

The only suggestion I have is to replace commercial names with the actual compounds for the anti-HE drugs listed on page 4 (Hepa-Merz.....)

Reviewer #3: The authors present an interesting crossover trial of AAC vs GS in nine patients with cirrhosis where oral AAC was associated with increased sleepiness on Karolinska but not with hypothesized pupillary light response. I have a few comments regarding the methodology.

1. Did you consider potential learning effects of repeated PHES and CRT? Typically, PHES is not repeated in such a short interval (i.e. 1-day) and as with any neuropsych evaluation there can be a learning effect. Summarizing the details of PHES results (i.e. each component of the test and score), by #day (1st vs 2nd) and by AAC vs GS can be helpful in understanding how the subjects performed after intervention. The fact that CRT time was reduced after AAC also suggests that the learning effects may have been an important factor.

2. Please include more details of liver disease in the subjects (MELD score, complications of liver cirrhosis, etc.). The heterogeneity of cirrhosis etiology in a small sample could have contributed to the negative result.

3. The figure 2 (schematic representation of pupillary light response) is difficult to read, either due to rendering or due to the font. Please revise to help with readability.

4. PSQI questions ask sleep quality over the past 1 month. How was this handled? One-time intervention is unlikely to affect the scoring over 1-month interval over 2-day study period.

6. PLOS authors have the option to publish the peer review history of their article (what does this mean?). If published, this will include your full peer review and any attached files.

Reviewer #1: **Yes: **Jeremy Franklin

Reviewer #2: **Yes: **Sara Montagnese

Reviewer #3: No

---

## [Author Response · Author response to Decision Letter 0]

15 Mar 2022

PLEASE NOTE, THAT OUR RESPOND TO REVIEWERS IS ATTACHED AS A FILE UNDER "ATTACH FILES". THE FOLLOWING IS A COPY OF THIS DOCUMENT. 

Journal Requirements:

Authors’ reply: 

Thank you for this comment. We have now edited the style of our paper to meet the requirements. 

2. Please ensure you have discussed any potential limitations of your study in the Discussion, including study design, sample size and/or potential confounders.

Authors’ reply: 

Thank you for this comment. In the discussion, we have described the following with regard to sample size and study design: “However, our sample size and lack of a control group of healthy subjects do not allow us to make firm conclusions on baseline PIPR levels in patients with cirrhosis.” 

3. Thank you for submitting your clinical trial to PLOS ONE and for providing the name of the registry and the registration number. The information in the registry entry suggests that your trial was registered after patient recruitment began. PLOS ONE strongly encourages authors to register all trials before recruiting the first participant in a study.

a. your reasons for your delay in registering this study (after enrolment of participants started);

b. confirmation that all related trials are registered by stating: “The authors confirm that all ongoing and related trials for this drug/intervention are registered”.

Authors’ reply: 

Thank you very much for this comment. We apologize for the delay. We have now inserted the following in the method section to meet both comment 3a and 3b: “The study was registered in clinicaltrials.gov. We did not register the study before the initiation of patient enrolment, as it was not a requirement in The Committee on Health Research Ethics for this type of national study with no drug intervention. We have no ongoing or related trials.”

(The project has been financially supported by a grant from the Novo Nordisk Foundation with grant no. NNF16OC0020490. The funders had no role in study design, data collection and analysis, decision to publish, or preparation of the manuscript.)

Authors’ reply: 

Thank you for this comment. We have now re-written the paragraph: 

“Anna Emilie Kann was supported by a pre-graduate scholarship from the Novo Nordisk Foundation (NNF16OC0020490). The funder had no role in study design, data collection, analysis, manuscript preparation, or decision to publish. No additional funders, either internally or externally.”

Authors’ reply: 

Thank you. We have now ensured that they are identical

Authors’ reply: 

Thank you for the comment. The ethics is written within the Method section: 

“The study was approved by The Committee on Health Research Ethics of the Copenhagen Capital Region, Denmark (Project ID: H15000210). All patients provided written informed consent and were not offered financial compensation. The study was conducted under the Helsinki Declaration.”

 

Review Comments to the Author

Reviewer #1: 

The main points of concern in this manuscript are:

(a) the rather confused description of the known and hypothesized causal relationships and pathways and the light shed on these by this study and 

(b) the incomplete adherence to standard methods for analysis of cross-over studies.

(a) The introduction explains that liver cirrhosis can cause elevated ammonia level (hyperammonaemia) and hepatic encephalopathy (HE), which in turn can lead to sleep disturbances. The effect on sleep could hypothetically be mediated by damage to retinal cells leading to disturbance of the circadian rhythm. The causal connections and their direction, either known or suspected, between hyperammonaemia, HE and sleep are not clearly described. 

Ammonia levels were challenged and measures of ammonia level, HE-related variables, eye function and sleep were performed and pair-wise associations between these measures were investigated. It would be useful if the authors could summarize which associations are supported by the study results and what these results tell us about the possible causal pathways.

(b) Analysis of results from a cross-over design should look for a period effect by comparing the results from the first and second periods, since there may be systematic differences due to the subjects’ experience in period 1. It is clear that with an odd number of subjects, the intervention (AAC/GS) is unbalanced with respect to period, which might potentially bias the AAC/GS comparison. With a whole week between periods, we can presumably take the carry-over effect from the first to second period to be negligible – if so, please state this assumption.

Further, where appropriate (e.g. PIPR, ammonia level, KSS) the measured values after intervention should be compared with the respective values before intervention (i.e. day 0 of the period), for instance by analyzing the changes (for each subject) during the period.

Authors’ reply to (a): 

Thank you for your comment. We have now rephrased the introduction to make it more clear what is known and hypothesized. 

We have now summarized the expected associations in the end of the introduction 

“Based on the above, we proposed a new theory not previously investigated. We hypothesized that hyperammonaemia lead to a dysfunction of the ipRGCs and thereby affect the regulation of sleep in patients with cirrhosis. We conducted a crossover trial to evaluate sleep and pupillary light response with pupillomety before and after an oral amino acid challenge. We hypothesized that an oral amino acid challenge induced hyperammonaemia, increased subjective sleepiness, worsened the psychometric tests for minimal encephalopathy, and impaired the pupillary response to blue light stimulation.”

and the beginning of the discussion: 

“Our study included nine patients with cirrhosis. Two-thirds had impaired self-reported sleep quality. The AAC intervention was associated with an acute increase in capillary ammonia concentration and subjective sleepiness but had no immediate effect on the melanopsin-mediated pupillary light response or the psychometric tests. We observed more heavy sleep the night after both experimental days compared to baseline sleep, but no independent effect of AAC on sleep.”

And in the conclusion:

“Our study showed that patients with cirrhosis had impaired sleep quality. An oral AAC resulted in an increase in the ammonia concentration and an associated subjective sleepiness within hours. We found no acute effect of the oral AAC on the melanopsin-mediated pupillary light response, and therefore we could not confirm our main hypothesis.” 

Authors’ reply to (b): 

Thank you for your comment. We agree. A cross over design was necessary, and by adding a one-week washout period, we presume that the effect of intervention is eliminated and would not affect the encephalopathy nor the sleeping. We have written the following in the method section: “To eliminate or reduce the carry-over effect, we separated the experimental days by one week.”

We primarily wanted to compare the differences between groups rather than within groups, although we agree that the change from baseline also is a relevant measure of the effect of the intervention. We believe the change from baseline is clearly assessable in our figures, and we have actually done the analysis of change from baseline for each type of measurement confirming a significant change from baseline to peak values in the AAC-group. We chose not to include the analysis since it does not change or add substantially to the conclusions.

Minor points

1. Definition of movement index (p.6): why is denominator time spent in bed, rather than TST? As defined, longer TST could lead to increased movement index.

Authors’ reply: 

Thank you for your comment. This was a typing mistake from our side. We have now corrected the denominator. The actual calculations were done by the proprietary ActiGraph software and was not influenced by this mistake. The following is the corrected calculation:

Movement index=(Time with movements during sleep (minutes))/(Total sleep time (minutes))×100

2. Definition of fragmentation index unclear (what is ‘Number of sleep epochs = 1 minute’?).

Authors’ reply: 

We have now rephrased the definition slightly:

“Fragmentation index=(Number of periods of one minute of sleep)/(Total sleep time (minutes))×100”

3. Please justify the assumed effect size (0.8 is unusually large). 

Authors’ reply:

We based the expectancy of a large effect size on the differences in PIPR seen in other studied groups of subjects: choroideremia vs. healthy subjects, where the ratio of the difference in means vs. standard deviation was approximately 0.8.

4. The distribution of baseline values before AAC and GS respectively should be compared, since similar baselines are need to ensure a fair comparison.

Authors’ reply:

We did compare the baseline values between AAC and GS in our primary analysis as reported in the manuscript. We observed no group differences at baseline.

5. In Fig. 3, are mean and standard deviation displayed? With only 9 subjects, it might be possible and more informative to plot the individual series.

Authors’ reply:

Thank you for your comment. In figure 3 and 4, the error bars represent the median absolute deviation, a robust statistics for non-normally distributed data or small series. The legibility of “spaghetti plots” of our data was not good, which is why we chose summary statistics instead of individual data series.

6. Table 2: IQR should be given as (lower quartile, upper quartile).

Authors’ reply:

Thank you for the suggestion. We have now changed the metric to 25% and 75% quartiles.

7. Throughout Results, exact p-values should be quoted.

Authors’ reply:

Thank you for pointing this out. We have now added exact p-values throughout the manuscript

8. The graphic quality of Fig. 2 is very low, to the point of illegibility.

Authors’ reply:

Thank you for your comment. We have now revised the figure. 

Reviewer #2: 

This is an interesting, innovative, well presented and honestly discussed piece of research.

The only suggestion I have is to replace commercial names with the actual compounds for the anti-HE drugs listed on page 4 (Hepa-Merz.....) 

Authors’ reply

Thank you for the suggestion. We have now changed the names: “lactulose, rifaximin, L-ornithine L-aspartate or branched chain amino acid supplementation”

Reviewer #3: 

The authors present an interesting crossover trial of AAC vs GS in nine patients with cirrhosis where oral AAC was associated with increased sleepiness on Karolinska but not with hypothesized pupillary light response. I have a few comments regarding the methodology.

1. Did you consider potential learning effects of repeated PHES and CRT? Typically, PHES is not repeated in such a short interval (i.e. 1-day) and as with any neuropsych evaluation there can be a learning effect. Summarizing the details of PHES results (i.e. each component of the test and score), by #day (1st vs 2nd) and by AAC vs GS can be helpful in understanding how the subjects performed after intervention. The fact that CRT time was reduced after AAC also suggests that the learning effects may have been an important factor.

Authors’ reply:

Thank you for your comment. We do agree this is relevant to consider. We tried to consider it by making a crossover design, even though it is not completely balanced. Further, we used unique PHES test for each test (4 in all). The reduced CRT index means an impairment of the CRT, which we interpret, could be due to tiredness and the ammonia increase. We have now mentioned the potential learning effect in the discussion: “The lack of association between hyperammonemia and PHES could be caused by a potential learning effect of the short period between tests. However, we tried to prevent this by applying four unique PHES test sets per patient.”

2. Please include more details of liver disease in the subjects (MELD score, complications of liver cirrhosis, etc.). The heterogeneity of cirrhosis etiology in a small sample could have contributed to the negative result.

Authors’ reply:

Thank you for your comment. We agree that the heterogeneity in the etiology could have contributed to the negative result, which we have now added in our discussion: “For example, the patients’ different liver etiology could have contributed to the negative result. Further, the population might have been more susceptible if the included patients had more severe liver disease, e.g., Child-Pugh class C and a history of hepatic encephalopathy. By this, the effect of intervention might have been more pronounced. In the design phase of the study, we decided to use the present inclusion criteria anticipating a rather high need for practical cooperation during the time-consuming assessments, in particular the pupillometry.”

Unfortunately, the components for the MELD and further complications to the cirrhosis is not available in our data set. Therefore, we cannot change table 1. Anyhow, Child Pugh Score is a good and reliable instrument to evaluation of the stage of liver disease 

3. The figure 2 (schematic representation of pupillary light response) is difficult to read, either due to rendering or due to the font. Please revise to help with readability. 

Authors’ reply:

Thank you for your comment. We have now revised the figure. 

4. PSQI questions ask sleep quality over the past 1 month. How was this handled? One-time intervention is unlikely to affect the scoring over 1-month interval over 2-day study period.

Authors’ reply:

Thank you for your comment. The PSQI was only completed once before the first experimental day. We have made this more clear in the method section: “Before the first experimental day, the patients completed the validated self-rated assessment tool Pittsburgh Sleep Quality Index (PSQI), which assesses sleep quality and disturbances within the preceding month.”

---

## [Decision Letter · Decision Letter 1]

21 Jul 2022

PONE-D-21-00133R1

The effect of induced hyperammonaemia on sleep and melanopsin-mediated pupillary light response in patients with liver cirrhosis: A single-blinded randomised crossover trial

PLOS ONE

Dear Dr. Kann,

Thank you for submitting your manuscript to PLOS ONE. After careful consideration, we feel that it has merit but does not fully meet PLOS ONE’s publication criteria as it currently stands. Therefore, we invite you to submit a revised version of the manuscript that addresses the points raised during the review process.

The manuscript has been evaluated by two reviewers, and their comments are available below.

One of the reviewers has raised a number of concerns. He requests improvements to the reporting of methodological aspects of the study, for example, regarding information on the timing of measured pupillary response, and intensity of the blue light stimulation.

Could you please carefully revise the manuscript to address all comments raised?

We look forward to receiving your revised manuscript.

Kind regards,

Lorena Verduci

Staff Editor

PLOS ONE

Journal Requirements:

Reviewers' comments:

Reviewer's Responses to Questions

**Comments to the Author**

1. If the authors have adequately addressed your comments raised in a previous round of review and you feel that this manuscript is now acceptable for publication, you may indicate that here to bypass the “Comments to the Author” section, enter your conflict of interest statement in the “Confidential to Editor” section, and submit your "Accept" recommendation.

Reviewer #2: All comments have been addressed

Reviewer #3: (No Response)

2. Is the manuscript technically sound, and do the data support the conclusions?

Reviewer #2: Yes

Reviewer #3: Yes

3. Has the statistical analysis been performed appropriately and rigorously? 

Reviewer #2: Yes

Reviewer #3: Yes

4. Have the authors made all data underlying the findings in their manuscript fully available?

Reviewer #2: Yes

Reviewer #3: Yes

5. Is the manuscript presented in an intelligible fashion and written in standard English?

Reviewer #2: Yes

Reviewer #3: Yes

6. Review Comments to the Author

Reviewer #2: There are no further comments from this reviewer - previous issues raised have been addressed adequately.

Reviewer #3: 1. The authors conducted an interesting, hypothesis-driven experiment with patients with a low-grade cirrhosis to test whether induced hyperammonemia acutely affect the pupillary response, as a surrogate measure of ipRGC function. There is a phase response curve to ipRGC; phase-shift effects response of ipRGC to the same intensity, spectrum, and duration of light will vary depending on the time of day. In addition, prior light history has been shown to affect the sensitivity of ipRGCs. Given the small sample size, it is not feasible to correct for multiple covariates. However, it will be useful to have additional information on the timing (both clock time and “circadian time” relative to the waketime of each subject) of measured pupillary response. It appears that the response was measured at hour 0, 2, and 4. Please include information on the clock/circadian time of these measures. Since ipRGC’s is most sensitive to light in the evening, the authors may consider including this in the discussion for a potential contributor to the lack of difference in the primary outcome.

2. Page 8. Pupilometer: What was the intensity of the blue light stimulation? Again, the response will depend on duration, intensity, and spectrum of the light.

3. Impaired sleep quality by which measure? PSQI? Please provide this information in the Table. (both baseline and after intervention)

4. Child-Pugh Score an outdated measure of cirrhosis severity because of the subjective scoring. Can you provide MELD or MELD-Na score for the patients?

5. The denominator of the Movement index should be TIB because TST does not include all of the “Time with movements during sleep”, which can be scored as wake during the sleep interval (therefore not included in TST). This likely won’t have a huge impact on the fragmentation index, but I wanted to make a note on it, disagreeing with the first reviewer.

6. Discussion paragraph #3: How do you define superficial sleep? Would it be based on PSG study? I would use caution defining deep vs superficial sleep based on actigraphy measures. The subjects had less WASO and higher sleep efficiency after the experiment, regardless the intervention arm. Therefore, I would conclude that they were exhausted from a whole day of activities, which might have resulted in a better sleep.

7. PLOS authors have the option to publish the peer review history of their article (what does this mean?). If published, this will include your full peer review and any attached files.

Reviewer #2: No

Reviewer #3: No

---

## [Author Response · Author response to Decision Letter 1]

17 Aug 2022

Response to reviewers

Journal Requirements

Authors’ reply: Thank you. We have ensured that the reference list is correct. We have not changed any current citations but have added an extra reference according to the reviewer's comments.

Review Comments to the Author

Reviewer #2: 

There are no further comments from this reviewer - previous issues raised have been addressed adequately.

Reviewer #3: 

1. The authors conducted an interesting, hypothesis-driven experiment with patients with a low-grade cirrhosis to test whether induced hyperammonemia acutely affect the pupillary response, as a surrogate measure of ipRGC function. There is a phase response curve to ipRGC; phase-shift effects response of ipRGC to the same intensity, spectrum, and duration of light will vary depending on the time of day. In addition, prior light history has been shown to affect the sensitivity of ipRGCs. Given the small sample size, it is not feasible to correct for multiple covariates. However, it will be useful to have additional information on the timing (both clock time and “circadian time” relative to the waketime of each subject) of measured pupillary response. It appears that the response was measured at hour 0, 2, and 4. Please include information on the clock/circadian time of these measures. Since ipRGC’s is most sensitive to light in the evening, the authors may consider including this in the discussion for a potential contributor to the lack of difference in the primary outcome.

Authors’ reply: Thank you for this important comment. We have now described it more thoroughly in the method section: “The study assessments were performed at different time points during the inclusion period, as shown in Table 1. The experimental days ran from morning (8.00 am) until afternoon (3.00 pm) to reduce differences in the circadian rhythm between patients.” And in the discussion, we have now included it as a potential contributor to the limited response: “In addition, the daytime assessment may have affected the results, as ipRGCs are more sensitive to light stimulation in the evening.” 

2. Page 8. Pupilometer: What was the intensity of the blue light stimulation? Again, the response will depend on duration, intensity, and spectrum of the light.

Authors’ reply: Thank you for this comment. We have now inserted more detailed information about pupilometry: “The duration of light stimulation was 20 seconds, and the illuminance level of both light colors was 100 lux.”

3. Impaired sleep quality by which measure? PSQI? Please provide this information in the Table. (both baseline and after intervention)

Authors’ reply: Thank you for this comment. We measured the baseline self-assessed sleep quality by PSQI, which was not repeated after the intervention, as described in the method section. Besides, we measure the sleep quality during the inclusion period by Actigraph. We have now inserted the number of patients with high vs. low PSQI scores in table 1. 

4. Child-Pugh Score an outdated measure of cirrhosis severity because of the subjective scoring. Can you provide MELD or MELD-Na score for the patients?

Authors’ reply: Unfortunately, the components for the MELD and further complications to the cirrhosis are not available in our data set. Therefore, we cannot change table 1. Anyhow, we consider the Child-Pugh Score a good and reliable instrument for evaluating the stage of liver disease. 

5. The denominator of the Movement index should be TIB because TST does not include all of the “Time with movements during sleep”, which can be scored as wake during the sleep interval (therefore not included in TST). This likely won’t have a huge impact on the fragmentation index, but I wanted to make a note on it, disagreeing with the first reviewer.

Authors’ reply: Thank you for your comment on this. We have used the definitions provided by ActiGraph, the producer of the sleep evaluation watch. We have considered your note but have decided to keep the definitions suggested by Actigraph to promote reproducibility. Please find definitions in this reference: ActiLife 6 Manual | ActiGraph (actigraphcorp.com), page 62. 

6. Discussion paragraph #3: How do you define superficial sleep? Would it be based on PSG study? I would use caution defining deep vs superficial sleep based on actigraphy measures. The subjects had less WASO and higher sleep efficiency after the experiment, regardless the intervention arm. Therefore, I would conclude that they were exhausted from a whole day of activities, which might have resulted in a better sleep.

Authors’ reply: Thank you for your comment. We have now changed the paragraph to clarify this: “We would have expected that sleep after AAC decreased sleep efficiency because Bersagliere et al. [4] found superficial sleep after an equivalent AAC intervention in a similar population. However, differences in the study design could explain the divergence because Bersagliere et al. assessed sleep during a nap in the same afternoon with EEG analysis. Moreover, heavier sleep in our population could be due to exhaustion from a long experimental day rather than the intervention itself.”

---

## [Decision Letter · Decision Letter 2]

12 Sep 2022

The effect of induced hyperammonaemia on sleep and melanopsin-mediated pupillary light response in patients with liver cirrhosis: A single-blinded randomized crossover trial

PONE-D-21-00133R2

Dear Dr. Kann,

We’re pleased to inform you that your manuscript has been judged scientifically suitable for publication and will be formally accepted for publication once it meets all outstanding technical requirements.

Kind regards,

Dario Ummarino, PhD

Senior Editor

PLOS ONE

Reviewers' comments:

Reviewer's Responses to Questions

**Comments to the Author**

1. If the authors have adequately addressed your comments raised in a previous round of review and you feel that this manuscript is now acceptable for publication, you may indicate that here to bypass the “Comments to the Author” section, enter your conflict of interest statement in the “Confidential to Editor” section, and submit your "Accept" recommendation.

Reviewer #2: All comments have been addressed

Reviewer #3: All comments have been addressed

2. Is the manuscript technically sound, and do the data support the conclusions?

Reviewer #2: Yes

Reviewer #3: Yes

3. Has the statistical analysis been performed appropriately and rigorously? 

Reviewer #2: Yes

Reviewer #3: Yes

4. Have the authors made all data underlying the findings in their manuscript fully available?

Reviewer #2: Yes

Reviewer #3: Yes

5. Is the manuscript presented in an intelligible fashion and written in standard English?

Reviewer #2: Yes

Reviewer #3: Yes

6. Review Comments to the Author

Reviewer #2: No further comments.

Reviewer #3: Thank you for addressing this reviewer's comments thoroughly. I have no additional comments and accept the manuscript in the current form.

7. PLOS authors have the option to publish the peer review history of their article (what does this mean?). If published, this will include your full peer review and any attached files.

Reviewer #2: **Yes: **Sara Montagnese

Reviewer #3: No

---

## [Editor Report · Acceptance letter]

19 Sep 2022

PONE-D-21-00133R2 

The effect of induced hyperammonaemia on sleep and melanopsin-mediated pupillary light response in patients with liver cirrhosis: a single-blinded randomized crossover trial 

Dear Dr. Kann:

I'm pleased to inform you that your manuscript has been deemed suitable for publication in PLOS ONE. Congratulations! Your manuscript is now with our production department. 

Kind regards, 

on behalf of

Dr Lorena Verduci 

%CORR_ED_EDITOR_ROLE%

PLOS ONE